# Identifying High-Risk Atrial Fibrillation in Diabetes: Evidence from Nomogram and Plasma Metabolomics Analysis

**DOI:** 10.3390/biomedicines13071557

**Published:** 2025-06-25

**Authors:** Qiushi Luo, Xiaozhu Ma, Shuai Mei, Qidamugai Wuyun, Li Zhou, Ziyang Cai, Yi Wen, Shitao Wang, Jiangtao Yan, Huaping Li, Jiahui Fan, Meiyan Dai

**Affiliations:** 1Division of Cardiology, Departments of Internal Medicine, Tongji Hospital, Tongji Medical College, Huazhong University of Science and Technology, Wuhan 430030, China; 2Hubei Key Laboratory of Genetics and Molecular Mechanisms of Cardiological Disorders, Wuhan 430030, China

**Keywords:** NHANES, liquid chromatography–mass spectrometry, integrated strategies, personalized management

## Abstract

**Background**: Diabetes significantly increases the risk of atrial fibrillation (AF), but identifying high-risk individuals remains a clinical challenge. This study aimed to improve AF risk stratification in diabetic patients through a combination of clinical modeling and untargeted metabolomic analysis. **Methods**: A clinical risk score was developed using data from the National Health and Nutrition Examination Survey (NHANES) and validated in an independent cohort from Tongji Hospital. Its association with long-term outcomes and its ability to predict AF recurrence after catheter ablation were assessed in follow-up studies. Additionally, untargeted plasma metabolomics was performed in a subset of diabetic patients with and without AF to explore underlying mechanism. **Results:** The risk score showed good predictive performance in both the development and validation cohorts and was significantly associated with clinical prognosis. When combined with left atrial diameter and AF type, it also improved the prediction of AF recurrence after ablation. Metabolomic profiling revealed notable disturbances in energy metabolism, heightened inflammatory activity, and elevated stress responses in AF patients, indicating a distinct metabolic risk profile. **Conclusions:** This study provided two approaches to identify high-risk AF in diabetic patients, discussed the underlying pathophysiological mechanisms, and compared their characteristics and applications. And integrated strategies could improve AF risk stratification and personalized management in the diabetic.

## 1. Introduction

Atrial fibrillation (AF) is a type of arrhythmia with a heavy health burden all over the world. Statistical results show that the incidence rate is approximately 3% [1]. AF is a multi-system-related disease. AF is associated with hypertension, coronary heart disease, structural heart disease, hyperthyroidism, chronic obstructive pulmonary disease (COPD), and advanced age. And diabetes is an endocrine disorder that also affects multiple systems. Common complications of diabetes include diabetic angiopathy, diabetic peripheral neuropathy, chronic kidney disease (CKD), and diabetic cardiomyopathy. It is worth noting that diabetes increases the risk of AF. The association between diabetes and AF is, on the one hand, related to comorbidities such as heart failure and hypertension, and on the other hand, involves molecular mechanisms related to inflammation, oxidative stress, and disturbances in glucose and lipid metabolism [2].

Currently, AF management follows a full-process, integrated, and multidisciplinary approach. As AF progresses with the development of the disease or the mutual promotion of comorbidities, some surgeries and medications may have suboptimal effects. Therefore, prevention, diagnosis, and treatment targeting the risk factors and high-risk characteristics of AF are crucial for effective AF management [3,4,5]. Furthermore, the liquid chromatography–mass spectrometry (LC–MS) approach offers an effective method for identifying low molecular weight substances that are not typically tested in routine hospital examinations and for identifying specific biomarkers for the management of AF and its comorbidities [6].

There are many existing studies on risk prediction models for AF, with models such as the C_2_HEST score being based on clinical characteristics [7]. There are also models developed using machine learning and multi-omics technologies; however, these models often face challenges in clinical application and widespread adoption due to the black-box nature of machine learning or the difficulties in collecting multi-omics data [8,9]. In addition, there is a lack of personalized management strategies aimed at identifying high-risk AF individuals within the diabetic population. Meanwhile, there is a lack of explanation and analysis of the comorbid relationships and pathophysiological mechanisms between AF and its risk factors, as well as a lack of summary and comparison of features from different modeling approaches.

The process of this study was as follows: (1) the AFDB score (the nomogram) was developed and tested for predicting AF risk in diabetic patients using NHANES data, with subsequent validation using data from Tongji Hospital. The association between the AFDB score and survival prognosis was also evaluated in the National Death Index (NDI) cohort, and its predictive value for AF recurrence after radiofrequency ablation was examined in diabetic patients at our hospital [10,11]; (2) untargeted plasma metabolomics analysis was performed on diabetic patients with and without AF at our hospital.

In conclusion, this study provided two approaches to identify high-risk AF in diabetic patients, discussed the underlying pathophysiological mechanisms, and compared their characteristics and applications.

## 2. Materials and Methods

### 2.1. Inclusion and Exclusion of the Population and Data Collection

The NHANES is a large, comprehensive, and representative research project aimed at assessing the health and nutritional status of U.S. adults and children. The website is: https://wwwn.cdc.gov/nchs/nhanes/Default.aspx (accessed on 1 December 2024).

First, in all NHANES datasets, following this search and inclusion-exclusion process, a total of 257 diabetic patients with AF from the 1999–2020 NHANES dataset were selected as the case group. In the 1999–2000, 2001–2002, and 2009–2010 datasets, 1683 diabetic patients without AF were included as the control group. The NHANES dataset population was used as the training set for score development. The variables involved in the inclusion and exclusion process of the NHANES, the variables used to collect basic information of the NHANES, and the variables in the NDI dataset were provided in the Appendix A, Table A1.

Additionally, a hospital data processing and application platform developed by Yidu Cloud Technology Co., Ltd. (Beijing, China) was used to search, include, and exclude subjects in case groups (diabetic patients with AF) and control groups (diabetic patients without AF) at Tongji Hospital, based on admission dates and discharge diagnoses. A total of 911 diabetic patients with AF from 2023 and 2819 diabetic patients without AF from December 2023 were included in the dataset. This part was used as the test dataset.

In the NHANES, we gathered relevant data for the corresponding population. Detailed definitions of the variables could be found on the NHANES website. For categorical data, we collected the following variables: gender, ethnicity, hypertension, infection, chronic bronchitis, hyperlipidemia, CKD, anemia, hyperthyroidism, coronary heart disease, stroke, hepatic insufficiency, and New York Heart Association II~IV (NYHA II~IV). For continuous data, we collected the following variables: age, body mass index (BMI), glycated hemoglobin (HbA1c), systolic and diastolic blood pressure, leucocytes, neutrophils, lymphocytes, platelets, C-reactive protein (CRP), triglycerides, low-density lipoprotein cholesterol (LDL-C), high-density lipoprotein cholesterol (HDL-C), total cholesterol (TC), creatinine, hemoglobin, free thyroxine (FT4), thyroid stimulating hormone (TSH), alanine aminotransferase (ALT), and N-terminal pro B-type natriuretic peptide (NT-proBNP). Mortality status and follow-up time were obtained from the NDI records linked to NHANES, available at: https://www.cdc.gov/nchs/data-linkage/mortality-public.htm (accessed on 1 December 2024). The missing data statistics for the training set were summarized in the Appendix A, Table A2 for reference.

Additionally, we used the Yidu Cloud platform to retrieve data from relevant populations at Tongji Hospital, Huazhong University of Science and Technology. Information such as admission date, gender, age, ethnicity, and discharge diagnosis were collected. As the data from 2023 was relatively recent, the medical records were complete, and no missing values were found in the test set for this study.

### 2.2. Statistical Analysis, Score Design, and Test

In the case of directly deleting missing values, we performed group comparisons of baseline data from the NHANES dataset. For continuous data, the correlation analysis between the two groups was based on the U test, while the correlation analysis for categorical data was based on the chi-square test. To facilitate regression analysis, we imputed the missing values in the population data before conducting logistic regression. For continuous data, missing values were imputed with the mean. For binary categorical data, missing values were imputed with 0.

A univariate logistic regression analysis was performed on the NHANES. Based on the results of the univariate logistic regression analysis, we removed overlapping variables between the laboratory tests represented by continuous data and the diagnoses or categorical data. We also removed variables with *p* ≥ 0.05 or odds ratio (OR) ≤ 1 in this stage. Afterward, we conducted a preliminary multivariate logistic regression analysis on the remaining variables. Variables whose OR direction in the preliminary multivariate logistic regression analysis differed from the univariate logistic regression analysis were excluded. Finally, the remaining variables were used for the final multivariate logistic regression analysis, and a nomogram for the AFDB score was generated based on the results. Collinearity analysis is provided in Appendix B, with a VIF > 10 considered suggestive of strong collinearity and taken into account as a reference during variable assessment.

Next, we continued to retrieve relevant information from the diabetes with AF and diabetes without AF on the application platform developed by Yidu Cloud (Beijing) Technology Co., Ltd. And then, using R, we calculated the scores and risk scores for both groups in the test set and compared the results with the actual outcomes. Additionally, we performed group difference analysis and univariate logistic regression analysis on this portion of the data.

Logistic regression analysis established a linear equation to describe the relationship between independent variables and log odds. The log odds were then converted to probabilities, which were used to predict and interpret outcome categories. The logistic regression model could be represented by the following equation:$$logit\left(p\right)=ln(\frac{p}{1-p}{)=\beta }_{0}+\beta_{1}X_{1}+\beta_{2}X_{2}+\cdots +\beta_{n}X_{n}\;n\geq 1$$
where $logit\left(p\right)$ represented the log odds, where *p* was the probability of the event occurring, $X_{1}$, $X_{2}$,…, $X_{n}$ represented the variables, $\beta_{0}$, $\beta_{1}$, $\beta_{2}$,…, $\beta_{n}$ represented the model parameters (coefficients). By estimating the parameter values, the best-fitting model can be obtained, which was used to predict the probability of the outcome variable.

The receiver operating characteristic (ROC) curve was a method for evaluating diagnostic performance. And the area under the curve (AUC) was also an indicator for assessing diagnostic effectiveness.

The confusion matrix was a method for evaluating diagnostic performance, with related metrics including true positive (TP), false positive (FP), true negative (TN), and false negative (FN). Among these, the Youden index was an indicator for assessing the authenticity of the model. In this study, the confusion matrix was based on the threshold corresponding to the maximum Youden index in the training set, which most accurately reflected the model’s generalizability. R 4.4.0 and SPSS Statistics 26 were used during these processes.

The calibration curve was used to assess the predictive accuracy of the model. Decision curve analysis (DCA) and clinical impact curve (CIC) were used to evaluate the clinical intervention benefits and the model’s practical clinical impact.

Meanwhile, we divided the subjects in NHANES into high-risk and low-risk groups based on the score median, and plotted the Kaplan–Meier (KM) curve for the two risk groups using the NDI dataset linked to this dataset. In April 2025, we conducted telephone follow-ups for some AF patients in the case group after radiofrequency ablation to record AF recurrence and when outcomes occurred.

### 2.3. Untargeted Metabolomics Analysis

From 1 September 2024, 10 diabetic patients with AF and 10 diabetic patients without AF were enrolled at Tongji Hospital, Huazhong University of Science and Technology, and plasma samples were collected for untargeted metabolomics analysis.

The metabolomics analysis was performed using a LC–MS system (Waters, Milford, MA, USA). In both the positive and negative ion modes, the mobile phase consisted of 0.1% formic acid in water and 0.1% formic acid in acetonitrile.

The Xevo G2-XS QTof mass spectrometer (Waters, Milford, Massachusetts, USA) acquired both primary and secondary mass spectra. Dual-channel data were collected at low and high collision energies (10–40 V) with a 0.2-s scanning frequency per spectrum. The ESI (Electrospray Ionization) ion source parameters included a capillary voltage of 2500 V (positive mode) or −2000 V (negative mode), a cone voltage of 40 V, an ion source temperature of 100 °C, desolvation gas temperature of 500 °C, backflush gas flow rate of 50 L/h, and desolvation gas flow rate of 800 L/h.

Raw data were processed using Progenesis QI software 2.4 (Waters, Milford, MA, USA) for peak extraction, alignment, and compound identification.

Principal component analysis (PCA) was performed on all metabolites to evaluate intergroup differences. In addition, orthogonal partial least squares (OPLS) analysis was conducted to assess the significance of group-specific features. A volcano plot of differentially altered metabolites (*p* < 0.05; VIP > 1) was generated. Pie charts and volcano plots were used to assess the classification and differential expression of metabolites. Pathway enrichment significance was evaluated using a hypergeometric distribution test based on the KEGG (Kyoto Encyclopedia of Genes and Genomes) database.

In addition, principal component analysis and correlation analysis were used to exclude samples with poor correlation, resulting in a final selection of 7 diabetic patients with AF and 6 diabetic patients without AF. The results were then reanalyzed.

## 3. Results

### 3.1. Study Flowchart

Figure 1 demonstrated the flowchart of this study. Figure 1a illustrated not only the inclusion and exclusion process for the nomogram training and testing sets, but also the sources of samples used for plasma metabolomics analysis. Figure 1b showed the sources of the follow-up cohorts for both the training and testing sets.

### 3.2. Baseline Characteristics

Table 1 demonstrated the basic information of the included population in the NHANES dataset. Table 2 is the univariate regression analysis of baseline data, variables, and outcomes in the Tongji Hospital. Table 3 is the baseline characteristics of cohort: recurrence in diabetic patients with AF after radiofrequency ablation (RFA) at Tongji hospital. The median follow-up time was 6 months in the recurrence group and 18 months in the non-recurrence group.

### 3.3. Variable Selections

Figure 2 demonstrated the variable selections based on the results of logistic regression analysis. Univariate regression analysis and preliminary multivariate logistic regression analysis identified the following variables for inclusion in the final multivariate regression analysis: age, male, non-Hispanic White, hypertension, chronic bronchitis, CKD, coronary heart disease, infection, NYHA II-IV, and hyperthyroidism. Collinearity analysis shown in Appendix B served as a reference, indicating that leukocytes, neutrophils, and lymphocytes exhibited notable collinearity in the univariate logistic regression.

### 3.4. Nomogram for the Score of Predicting the Risk of AF in the Diabetic

As shown in Figure 3, based on the final multivariate regression analysis, a nomogram to identify high-risk AF in diabetes was constructed. The final included variables could be used to calculate the total points and risk score.

### 3.5. The Evaluation for the AFDB Score for the NHANES and the Tongji Hospital

Figure 4 demonstrated the evaluation for the AFDB score for the NHANES and the Tongji Hospital. Figure 4a displayed the ROC curve for the AFDB score based on the NHANES database, with an AUC of 0.85. Figure 4b–e represented the calibration curve, DCA curve, confusion matrix, and CIC curve of the train dataset, respectively. The DCA curve indicated that the AFDB score provided good clinical net benefit in the training set. The calibration curve closely aligned with the ideal reference line, suggesting that the model has high predictive accuracy. Figure 4f showed the KM curve for survival prognosis based on the AFDB score for the NHANES dataset, distinguishing high-risk and low-risk populations based on the median score (The median score was 90.72). It could be observed that the high-risk population had significantly worse survival prognosis compared to the low-risk group (*p* < 0.001). Figure 4g showed the ROC curve for the AFDB score based on the data from Tongji Hospital, with an AUC of 0.78. Figure 4h–k represented the calibration curve, DCA curve, confusion matrix, and CIC curve of the test dataset, respectively. The DCA curve indicated that the AFDB score provided a certain level of clinical net benefit in the testing set. The calibration curve’s alignment with the ideal reference line suggests that the model has a certain degree of predictive accuracy. However, compared to the results in the training set, the model performed less favorably, indicating that its generalizability in external testing still has room for improvement. Figure 4l showed the AFDB score’s predictive value for post-RFA recurrence in the Tongji hospital follow-up cohort, with the combination of left atrial diameter and preoperative AF type improving the prediction accuracy to 0.72.

### 3.6. Plasma Metabolomics Analysis

Table 4 demonstrated the baseline data before and after excluding confounding subjects. Figure 5 and Figure 6 illustrated the metabolic differences between the case and control groups before and after excluding confounding subjects. Figure 5a showed that there were some confounding subjects between the two groups. After excluding these subjects, seven subjects remained in the case group and six subjects remained in the control group. Figure 6f presented the KEGG enrichment analysis after exclusion. A summary of the top 10 metabolic pathways revealed three key features in the case group: significant disturbances in energy metabolism (branched-chain amino acid biosynthesis and degradation pathways and the tricarboxylic acid cycle), increased inflammatory activity (arachidonic acid metabolism and linoleic acid metabolism), and elevated stress responses (steroid metabolism). Figure 7a represented the specific metabolites involved in the feature of significant disturbances in energy metabolism. Figure 7b showed one representative metabolite, isoleucine, which indicated the accumulation of branched-chain amino acid with disrupted energy utilization. Figure 7c represented the specific metabolites related to increased inflammatory activity. Figure 7d showed one representative metabolite, thromboxane B2, indicating activation of the arachidonic acid cyclooxygenase (COX) pathway in the inflammatory response. Figure 7e summarized the specific metabolites involved in elevated stress responses. Figure 7f showed one representative metabolite, cortisone, represented adrenal cortex hyperfunction under stress conditions.

## 4. Discussion

This study provided two approaches to identify high-risk AF populations among diabetic patients. (1) This study developed a nomogram for identifying high-risk AF in diabetes, which underwent cross-ethnic validation and was supplemented with follow-up cohorts. The nomogram incorporated variables such as age, sex, ethnicity, hypertension, chronic bronchitis, chronic kidney disease (CKD), coronary heart disease, infection, New York Heart Association (NYHA) classification, and hyperthyroidism. It outperformed the similarly structured C_2_HEST score by an average of five points. The nomogram is intuitive, simple, and highly interpretable, making it easy to implement in clinical practice, with a particular emphasis on comorbidity management [7]. (2) Plasma metabolomics analysis identified three key characteristics associated with high risk of atrial fibrillation in diabetes: more significant disruption of energy utilization, more severe inflammation, and stronger stress responses, which are closely related to the pathophysiology of both diabetes and AF.

Numerous studies have developed risk scores for predicting AF risk [9]. Meanwhile, Schnabel et al., Chamberlai et al., FIND-AF model, The CHARGE-AF score identified a lot of clinical factors associated with AF occurrence [8,12,13,14]. However, these models had several limitations, including reliance on single-center or single-source data, absence of cross-ethnic external validation, lack of comparative analysis between nomogram and metabolomics approaches, dependence on complex algorithms with limited interpretability, and restricted generalizability. Notably, none of these models were specifically developed for diabetic populations. It was noteworthy that the C_2_HEST score, similar to our model in terms of simplicity and interpretability [7], which performed worse than ours. This performance gap might be attributed to the C_2_HEST score’s lack of specificity for diabetic populations. This performance gap might be attributed to the C_2_HEST score’s lack of specificity for diabetic populations. Meanwhile, our scoring system drew on diverse datasets from North America to Asia, including both U.S. and Chinese populations, which complemented each other and demonstrated a broad application range and strong generalizability.

The latest guideline not only emphasized the importance of preventing AF before its electrocardiographic confirmation by introducing the concepts of AF stage 1 and stage 2, but also highlighted that, in addition to rate control and restoration of sinus rhythm, managing risk factors and addressing atrial structural remodeling were crucial for the prevention and treatment of AF [5]. The nomogram in this study included variables such as age, sex, ethnicity, hypertension, CKD, and coronary heart disease. While factors like age and sex are non-modifiable, the remaining conditions are treatable. Hypertension, CKD, heart failure and coronary heart disease are common diabetes complications [5]. In other words, lowering blood pressure, improving heart failure, treating coronary heart disease, controlling infections (or pulmonary infection), improving renal function, controlling COPD (or chronic bronchitis), and managing hyperthyroidism were crucial for the prevention of AF. Furthermore, diabetic subjects with higher AFDB scores (high-risk diabetic population for AF) had a higher risk of mortality (*p* < 0.01) and shorter follow-up time (*p* < 0.01). Therefore, preventing and managing AF in diabetic patients was of great importance for improving survival and prognosis, which suggested related measures could improve prognosis and prolong survival in the diabetic. In addition, this score also provided certain guidance for predicting the recurrence of AF after surgery.

It should be emphasized that both diabetes and AF are highly heterogeneous diseases. Diabetes not only includes type 1 and type 2 diabetes, but also broadly encompasses gestational diabetes, mitochondrial diabetes, and diabetes secondary to exocrine pancreatic disorders [15]. Similarly, AF can be classified based on episode characteristics and duration into paroxysmal, persistent, and permanent AF, and can also be divided etiologically into valvular and non-valvular AF [5]. Given the significant variability in individual pathophysiological mechanisms, elucidating the disease mechanisms on a personalized basis is crucial for understanding the onset and progression of these conditions.

Wang et al. reviewed the specific mechanisms underlying the increased incidence of AF in patients with diabetes. On the one hand, these mechanisms include glycemic variability, inflammatory responses, and oxidative stress; on the other hand, they involve structural remodeling of the heart, electromechanical remodeling, electrical remodeling, and autonomic remodeling [2]. Firstly, both the arachidonic acid metabolism pathway and its upstream linoleic acid metabolism pathway, which were identified in the enrichment analysis, are involved in inflammatory responses [16]. In addition, the elevated plasma levels of cortisol served as evidence of stress in the case group. Moreover, increased levels of various steroid hormones other than cortisol also suggested adrenal cortex hyperfunction, reflecting both stress and autonomic remodeling [17]. Furthermore, as the heart is a highly energy-demanding organ, diabetic patients often experience impaired cardiac energy metabolism. Hahn et al. conducted a metabolomic analysis using plasma and myocardial samples from patients with heart failure with preserved ejection fraction (HFpEF). The metabolic profile they summarized—“HFpEF is a syndrome with substantial fuel inflexibility”—is highly consistent with the findings in our metabolomic analysis [18]. Notably, both AF and diabetes are major comorbidities of HfpEF [19,20]. Thus, this substantial fuel inflexibility also exists in high-risk AF patients with diabetes, characterized by the accumulation of long-chain acylcarnitines, increased branched-chain amino acids, buildup of tricarboxylic acid cycle intermediates, disrupted ATP metabolic homeostasis, and overall mitochondrial dysfunction.

Therefore, the metabolic characteristics of AF in diabetes summarized in this study, which referred to more significant disruption of energy utilization, inflammation, and stress responses, align with the underlying pathophysiology of both diabetes and AF observed in previous studies. These molecular-level mechanistic features are often present across different clinical phenotypes of diabetes and AF. Meanwhile, numerous studies summarized the characteristics of metabolomics, proteomics, genomics, and transcriptomics in AF or diabetes [21,22,23,24]. Integrating multi-omics data (with a focus on plasma metabolomics in this study) greatly aids our understanding of the mechanisms underlying high-risk atrial fibrillation in diabetes.

Precisely because of the heterogeneity of diabetes and AF, this study did not use “black-box” artificial intelligence algorithms in the modeling process of the Electronic Health Record-integrated (EHR-integrated) tool [25,26]. Instead, variable selection was based on traditional *p*-values and odds ratios (OR), and modeling was performed using logistic regression. Additionally, results from plasma metabolomics of the case and control groups from our center were incorporated to help explain potential pathogenic mechanisms. The reason for this approach lies in fully considering the target users of the tool—clinicians. In practical applications, clinical prediction models often face issues of overfitting and underfitting [27]. Overfitting occurs when the model fits the training data too closely but has poor generalizability. Compared with artificial intelligence models, ours is more transparent, so if overfitting occurs, users (especially clinicians) can interpret and adjust the AFDB nomogram based on pathophysiological knowledge, characteristics of comorbidities in their own patients, and plasma metabolomic profiles. Underfitting occurs when the model does not fit the training data adequately and fails to capture underlying patterns. Since the modeling in this study used NHANES, a publicly available and transparent database, if underfitting arises, users (especially clinicians) can expand the AFDB nomogram training set using their own local data or multi-omics data (such as the metabolomics data in this study) to improve the model’s performance across various scenarios.

On the one hand, the advantage of the EHR-integrated tool is that it is derived from clinical data, making it relatively easy to obtain and apply. However, its limitation lies in the presence of bias when applied to different populations. In the NHANES dataset in this study, the variable with the highest OR was chronic bronchitis; while in the validation data from Tongji Hospital, the highest OR was for NYHA II–IV. This discrepancy is due to admission rate bias, which also affected the performance of the AFDB score in our hospital. On the other hand, plasma metabolomics offers a deeper mechanistic understanding, facilitating individualized interpretation of disease characteristics in different populations. Its drawbacks are that it is not routinely performed in clinical laboratories, has relatively high costs, and it is difficult for a single center to obtain large-sample data.

Yao et al. demonstrated that combining clinical and polygenic risk scores yielded the highest predictive accuracy for AF [9]. Therefore, the comprehensive integration and interpretation of different types of data will promote personalized management and precision treatment of diabetes with AF.

It is worth noting that the sample size and its representativeness often influence the reliability of conclusions. First, Gumprecht et al. conducted a prospective study and found that the incidence of AF among diabetic patients in the Polish population was approximately 25% [28]. The AFDB score includes 10 independent variables, and based on the Events Per Variable (EPV) method for sample size estimation (with EPV set at 10 and the training set proportion at 0.52), at least 400 cases are needed in the training set. Considering a 10% rate of invalid samples, a total sample size of at least 855 cases is required. The sample size used in this part of our study fully meets this requirement. In addition, we used G*Power 3.1 (http://www.gpower.hhu.de) to calculate the classification power of the plasma metabolites shown in Figure 7 (n = 7 AF cases vs. n = 6 controls) [29]. The statistical method used was a two-tailed *t*-test for two independent groups with a significance level of *p* = 0.05. The minimum classification powers for the three representative metabolites involved in the mechanisms shown in Figure 7b,e,f—namely isoleucine, TXB2, and cortisone—were 1.00, 1.00, and 0.91, respectively, all exceeding the threshold of 0.80. Appendix C shows the means and standard deviations of these three metabolites in the two groups. Future studies require larger, multicenter cohorts.

Diabetes combined with atrial fibrillation is often associated with poor treatment outcomes [30,31]. This study provided two approaches to identify high-risk AF in diabetic patients, discussed the underlying pathophysiological mechanisms, and compared their characteristics and applications. And integrated strategies could improve AF risk stratification and personalized management in the diabetic.

## Figures and Tables

**Figure 1 biomedicines-13-01557-f001:**
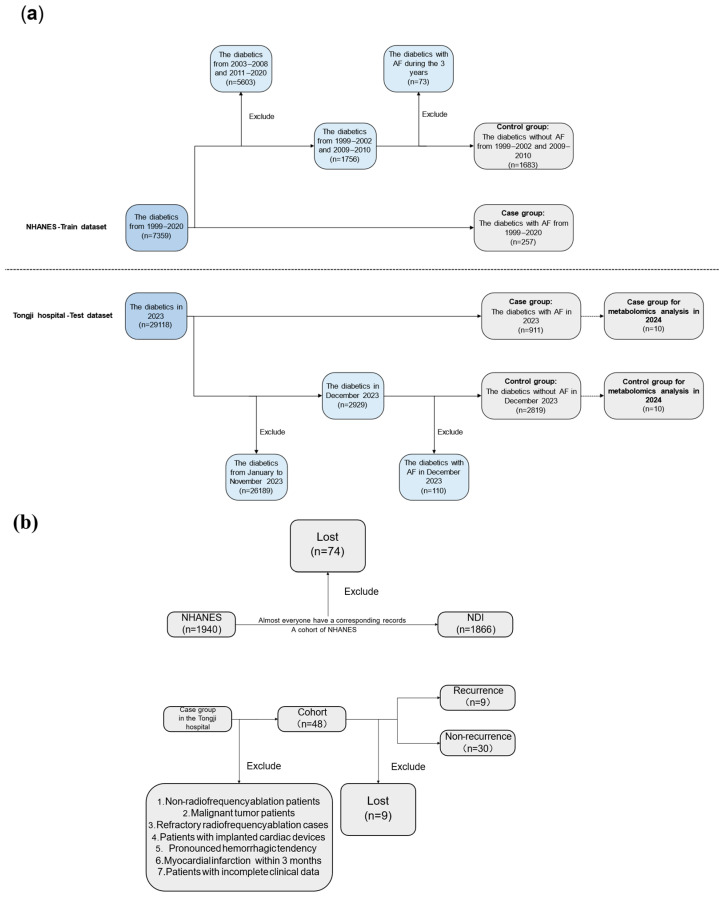
Study flowchart. (**a**) Enrollment and exclusion process of the training and testing populations for the nomogram. (**b**) Enrollment and exclusion process of the cohort. Note: NHANES, National Health and Nutrition Examination Survey; NDI, National Death Index; AF, atrial fibrillation.

**Figure 2 biomedicines-13-01557-f002:**
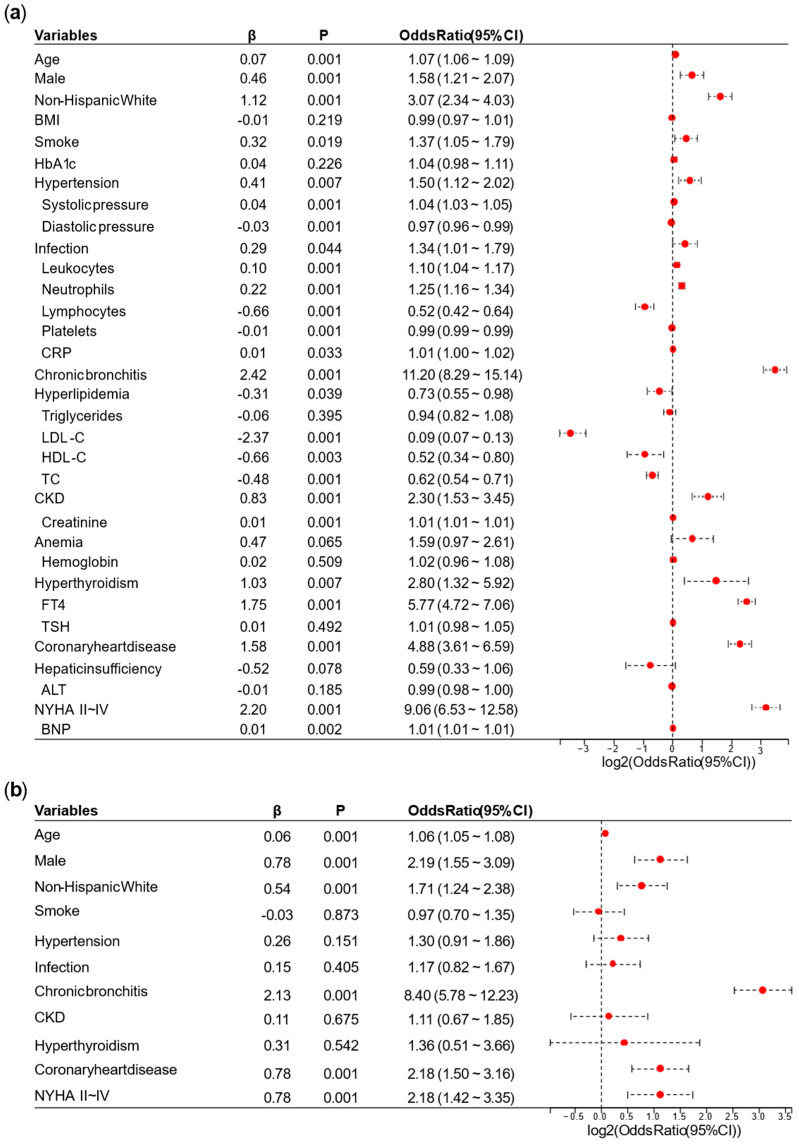
Variable selections based on the results of logistic regression analysis. (**a**) Univariate logistic regression analysis. (**b**) Preliminary logistic regression analysis. (**c**) Final logistic regression analysis. Note: BMI, body mass index; CRP, C-reactive protein; LDL-C, low—density lipoprotein cholesterol; HDL-C, high-density lipoprotein cholesterol; TC, total cholesterol; CKD, chronic kidney disease; FT4, free thyroxine 4; TSH, thyroid-stimulating hormone; ALT, alanine aminotransferase; NYHA II~IV, New York Heart Association II~IV; BNP, B-type natriuretic peptide. The red dots and dashed lines show the confidence intervals of ORs.

**Figure 3 biomedicines-13-01557-f003:**
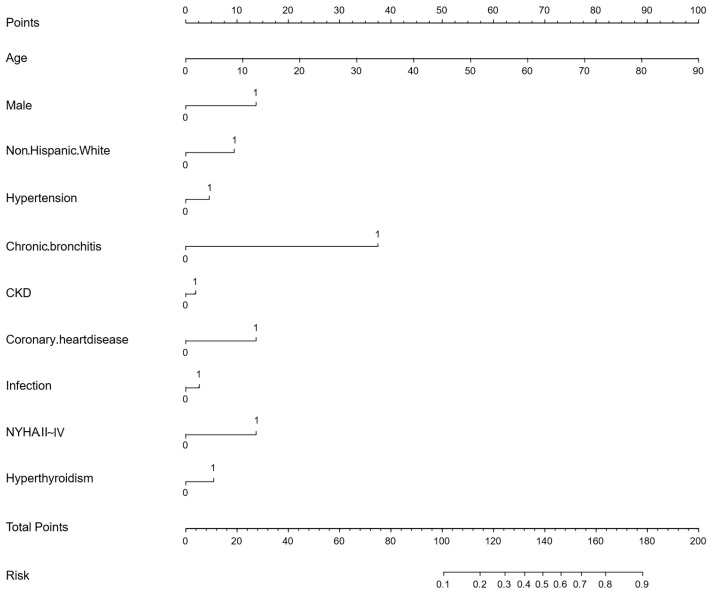
Nomogram for the score of predicting the risk of AF in the diabetic. Note: CKD, chronic kidney disease; NYHA II~IV, New York Heart Association II~IV.

**Figure 4 biomedicines-13-01557-f004:**
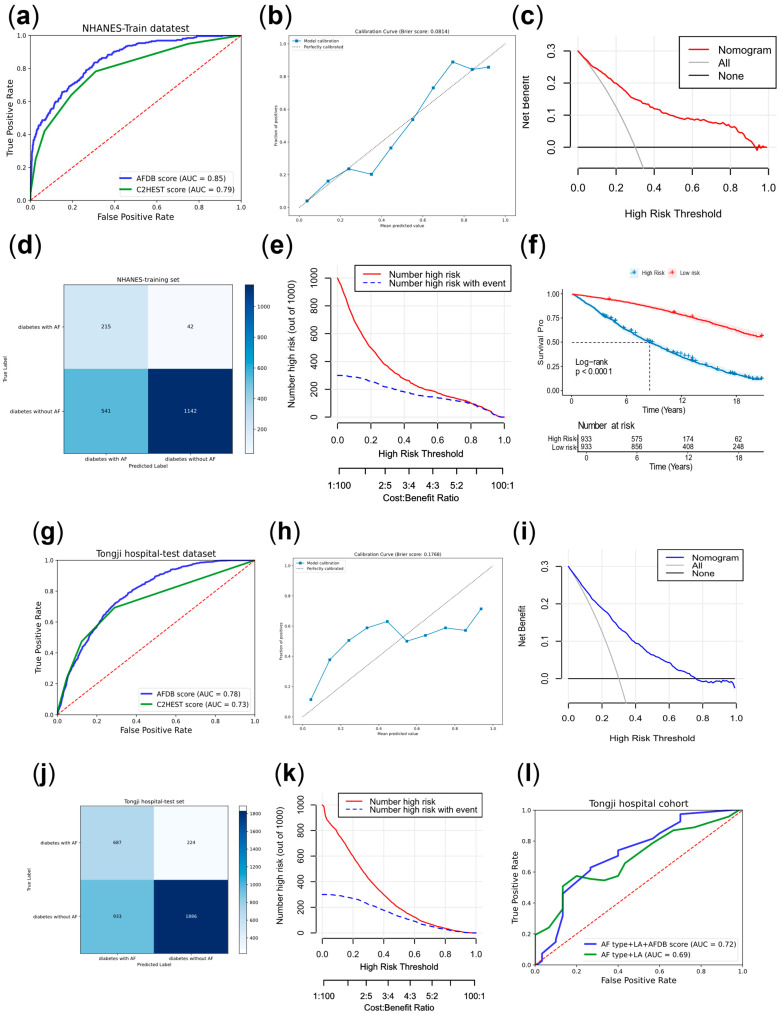
The evaluation for the AFDB score for the NHANES database and the Tongji Hospital. (**a**) ROC in the train dataset. (**b**) Calibration curve in the train dataset. (**c**) DCA in the train dataset. (**d**) Confusion matrix in the train dataset. (**e**) CIC in the train dataset. (**f**) KM plot of the risk score in the train dataset. (**g**) ROC in the test dataset. (**h**) Calibration curve in the test dataset. (**i**) DCA in the test dataset. (**j**) Confusion matrix in the test dataset. (**k**) CIC in the test dataset. (**l**) ROC in the test dataset. Note: NHANES, National Health and Nutrition Examination Survey; KM, Kaplan–Meier; ROC, receiver operating characteristic; AUC, area under the ROC curve; DCA, decision curve analysis; CIC, clinical impact curve.

**Figure 5 biomedicines-13-01557-f005:**
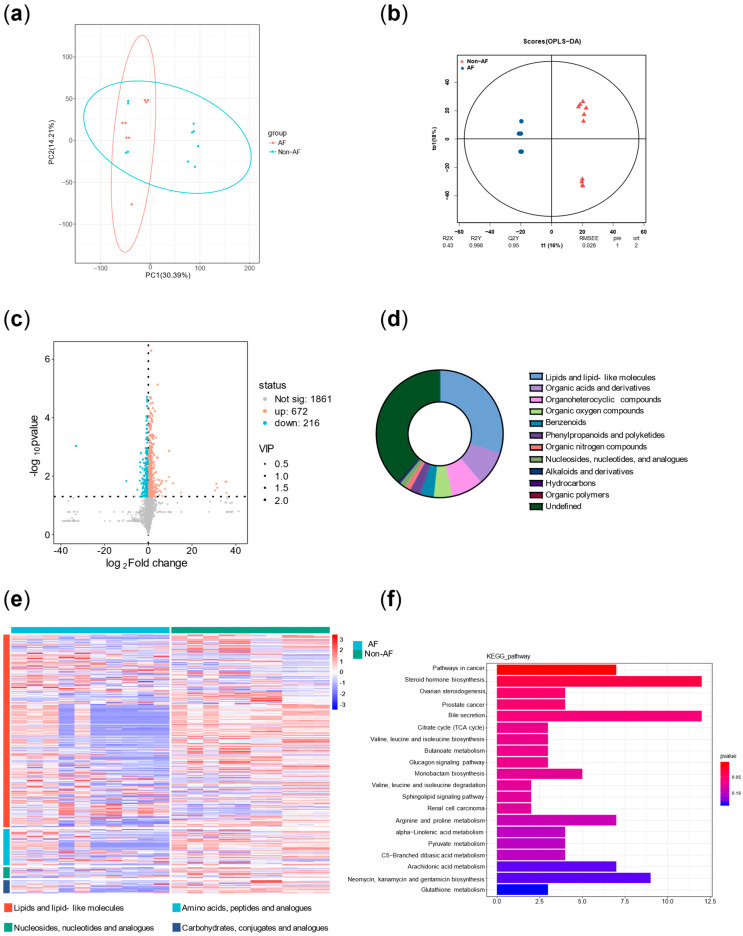
Overview of the metabolisms difference of the two groups before excluding confounding subjects. (**a**) PCA between two groups before excluding confounding subjects. (**b**) OP-LS before excluding confounding subjects. (**c**) Volcano plot before excluding confounding subjects. (**d**) Pie chart before excluding confounding subjects. (**e**) Heatmap before excluding confounding subjects. (**f**) KEGG enrichment analysis before excluding confounding subjects. Note: AF, atrial fibrillation; PCA, principal component analysis; OP-LS, orthogonal partial least squares; KEGG, Kyoto Encyclopedia of Genes and Genomes.

**Figure 6 biomedicines-13-01557-f006:**
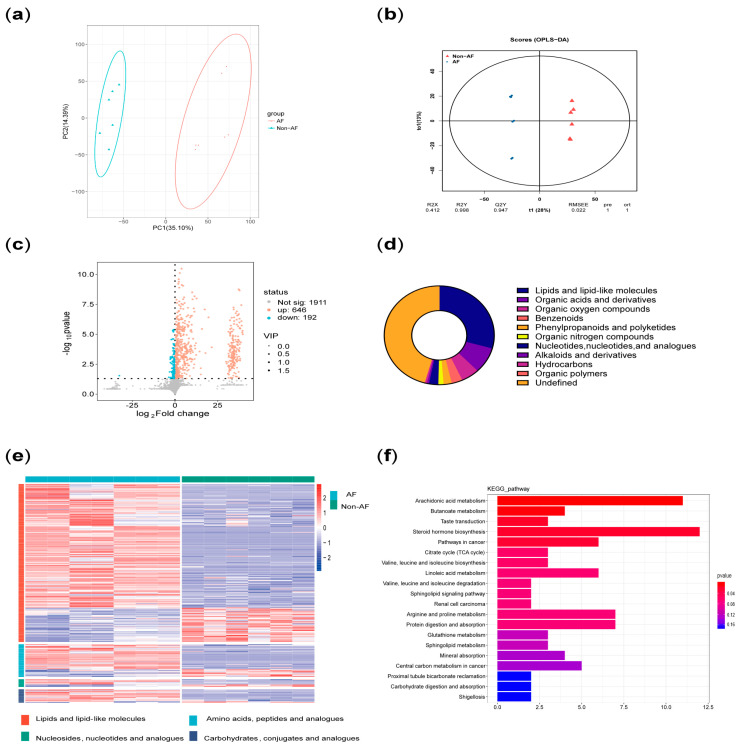
Overview of the metabolisms difference of the two groups after excluding confounding subjects. (**a**) PCA after excluding confounding subjects. (**b**) OP-LS after excluding confounding subjects. (**c**) Volcano plot after excluding confounding subjects. (**d**) Pie chart after excluding confounding subjects. (**e**) Heatmap after excluding confounding subjects. (**f**) KEGG enrichment analysis after excluding confounding subjects. Note: AF, atrial fibrillation; PCA, principal component analysis; OP-LS, orthogonal partial least squares; KEGG, Kyoto Encyclopedia of Genes and Genomes.

**Figure 7 biomedicines-13-01557-f007:**
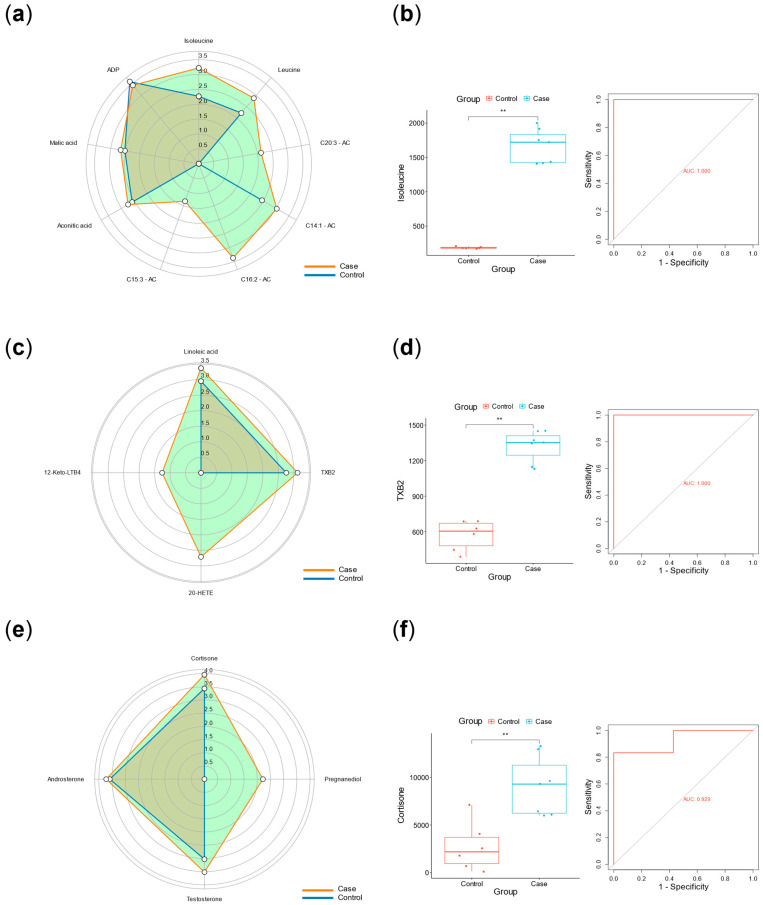
The plasma metabolic characteristics of high-risk AF in diabetes. (**a**) Radar plot of energy metabolism disruption. (**b**) Differences in the relative content of isoleucine and ROC curve. (**c**) Radar plot of inflammatory response. (**d**) Differences in the relative content of TXB2 and ROC curve. (**e**) Radar plot of corticosteroids, indicating a stronger stress response. (**f**) Differences in the relative content of cortisol and ROC curve. Note: AC, acylcarnitine; TXB2, thromboxane B2; HETE, hydroxyeicosatetraenoic acid; LTB4, leukotriene B4. The data in the radar chart were log10 transformed. ** *p* < 0.01.

**Table 1 biomedicines-13-01557-t001:** Basic information of the included population in the NHANES dataset.

Variables	Total (n = 1940)	Control (n = 1683)	Case (n = 257)	*p*
Age, years	62.83 ± 14.92	61.45 ± 15.16	71.85 ± 9.00	**<0.001**
Male (%)	988(50.93)	832(49.4)	156(60.7)	**<0.001**
Non-Hispanic white (%)	763 (39.33)	601 (35.71)	162 (63.04)	**<0.001**
BMI, kg/m^2^	31.61 ± 7.27	31.68 ± 7.29	31.12 ± 7.14	0.236
Smoke (%)	954 (50.24)	810 (49.33)	144 (56.03)	**0.046**
HbA1c, %	7.32 ± 2.16	7.30 ± 2.23	7.47 ± 1.60	0.227
Hypertension (%)	1281 (67.10)	1092 (66.06)	189 (73.83)	**0.014**
Systolic pressure, mmHg	121.26 ± 21.20	119.44 ± 20.48	133.89 ± 21.86	**<0.001**
Diastolic pressure, mmHg	66.75 ± 13.59	67.16 ± 13.65	63.85 ± 12.82	**0.002**
Infection (%)	499 (28.93)	419 (27.71)	80 (37.56)	**0.003**
Leukocytes, 1000 cells/μL	7.59 ± 2.26	7.53 ± 2.22	8.00 ± 2.49	**0.005**
Neutrophils, 1000 cells/μL	4.59 ± 1.79	4.51 ± 1.72	5.23 ± 2.13	**<0.001**
Lymphocytes, 1000 cells/μL	2.15 ± 0.97	2.20 ± 0.94	1.86 ± 1.07	**<0.001**
Platelets, 1000 cells/μL	245.06 ± 73.22	248.46 ± 73.16	221.29 ± 69.27	**<0.001**
CRP, mg/L	6.45 ± 12.61	6.25 ± 12.59	8.15 ± 12.71	**0.019**
Chronic bronchitis (%)	262 (13.88)	135 (8.24)	127 (50.80)	**<0.001**
Hyperlipidemia (%)	638 (38.18)	568 (38.90)	70 (33.18)	0.109
Triglycerides, mmol/L	2.02 ± 1.84	2.03 ± 1.56	1.96 ± 3.04	**0.007**
LDL-C, mmol/L	2.73 ± 0.90	2.81 ± 0.89	2.25 ± 0.82	**<0.001**
HDL-C, mmol/L	1.23 ± 0.36	1.24 ± 0.36	1.17 ± 0.35	**0.006**
TC, mmol/L	4.95 ± 1.28	5.02 ± 1.22	4.50 ± 1.58	**<0.001**
CKD (%)	143 (8.66)	108 (7.45)	35 (17.33)	**<0.001**
Creatinine, µmol/L	88.43 ± 76.61	85.75 ± 77.11	108.42 ± 69.69	**<0.001**
Anemia (%)	110 (6.39)	89 (5.89)	21 (9.91)	**0.025**
Hemoglobin, g/dL	13.46 ± 2.56	13.45 ± 2.66	13.56 ± 1.58	0.143
Hyperthyroidism (%)	34 (12.98)	24 (10.57)	10 (28.57)	**0.007**
FT4, pmol/L	11.11 ± 2.78	10.91 ± 2.69	12.41 ± 3.03	**0.002**
TSH, μIU/mL	2.26 ± 5.23	2.26 ± 5.40	2.39 ± 1.52	**0.018**
Coronary heart disease (%)	261 (13.95)	170 (10.46)	91 (36.99)	**<0.001**
Hepatic insufficiency (%)	152 (9.20)	139 (9.59)	13 (6.44)	0.147
ALT, U/L	25.18 ± 17.16	25.36 ± 17.76	23.88 ± 11.92	0.966
NYHA II~IV (%)	187 (9.72)	96 (5.73)	91 (36.40)	**<0.001**
NT-proBNP, pg/mL	537.17 ± 2503.85	498.78 ± 2524.41	1076.15 ± 2143.78	**<0.001**
Stroke (%)	203 (10.71)	155 (9.46)	48 (18.75)	**<0.001**
Mortality status (%)	933 (50.00)	771 (46.73)	162 (75.00)	**<0.001**
Follow-up time, years	129.72 ± 67.94	132.70 ± 66.70	92.16 ± 72.27	**<0.001**

Note: NHANES, National Health and Nutrition Examination Survey; BMI, body mass index; CRP, C-reactive protein; LDL-C, low-density lipoprotein cholesterol; HDL-C, high-density lipoprotein cholesterol; TC, total cholesterol; CKD, chronic kidney disease; FT4, free thyroxine 4; TSH, thyroid-stimulating hormone; ALT, alanine aminotransferase; NYHA II~IV, New York Heart Association II~IV; NT-proBNP, N-terminal pro-B-type natriuretic peptide. The bold value indicates statistically significant difference.

**Table 2 biomedicines-13-01557-t002:** Baseline characteristics at Tongji Hospital and univariate regression analysis of specific variables.

Variables	Control (n = 2819)	Case (n = 911)	*p*	OR (95%CI)	*p*(OR)
Age, years	57.85 ± 15.31	71.79 ± 9.93	**<0.001**	**1.09 (1.08~1.10)**	**<0.001**
Male (%)	1557 (55.23)	556 (61.03)	**0.002**	**1.27 (1.09~1.48)**	**0.002**
Han Chinese (%)	2803 (99.43)	907 (99.56)	0.841	0.77 (0.26~2.32)	0.645
BMI, kg/m^2^	23.66 ± 12.47	24.87 ± 4.44	**<0.001**		
Smoke (%)	252 (9.12)	122 (13.45)	**<0.001**		
HbA1c, %	8.40 ± 5.04	7.50 ± 1.58	**<0.001**		
Hypertension (%)	1302 (46.19)	554 (60.81)	**<0.001**	**1.81 (1.55~2.11)**	**<0.001**
Systolic pressure, mmHg	127.86 ± 41.48	137.35 ± 344.38	0.410		
Diastolic pressure, mmHg	78.35 ± 15.78	103.23 ± 29.82	**<0.001**		
Pulmonary infection (%)	331 (11.74)	211 (23.16)	**<0.001**	**2.27 (1.87~2.75)**	**<0.001**
Leukocytes, 1000 cells/μL	7.26 ± 3.21	7.45 ± 4.56	0.295		
Neutrophils, 1000 cells/μL	5.02 ± 2.99	5.32 ± 4.19	0.753		
Lymphocytes, 1000 cells/μL	1.57 ± 0.72	1.40 ± 1.06	**<0.001**		
Platelets, 1000 cells/μL	226.32 ± 415.24	195.05 ± 80.46	**<0.001**		
CRP, mg/L	18.98 ± 41.61	24.79 ± 51.03	**0.028**		
Chronic bronchitis (%)	49 (1.74)	45 (4.94)	**<0.001**	**2.94 (1.95~4.43)**	**<0.001**
Hyperlipidemia (%)	723 (26.21)	155 (17.09)	**<0.001**		
Triglycerides, mmol/L	2.92 ± 11.46	1.71 ± 1.22	**<0.001**		
LDL-C, mmol/L	2.80 ± 9.23	2.08 ± 0.84	**<0.001**		
HDL-C, mmol/L	1.10 ± 0.45	0.98 ± 0.30	**<0.001**		
TC, mmol/L	4.68 ± 10.85	3.66 ± 1.06	**<0.001**		
CKD (%)	551 (19.55)	344 (37.76)	**<0.001**	**2.50 (2.12~2.94)**	**<0.001**
Creatinine, µmol/L	115.38 ± 153.62	124.51 ± 140.24	0.122		
Anemia (%)	109 (12.02)	422 (15.27)	**0.008**		
Hemoglobin, g/L	130.85 ± 231.79	123.64 ± 26.77	0.489		
Hyperthyroidism (%)	29 (1.03)	23 (2.52)	**<0.001**	**2.49 (1.43~4.33)**	**0.001**
FT4, pmol/L	14.92 ± 35.40	16.79 ± 8.89	**<** **0.001**		
TSH, μIU/mL	2.36 ± 3.99	3.09 ± 3.96	**0.009**		
Coronary heart disease (%)	558 (19.79)	370 (40.61)	**<0.001**	**2.77 (2.36~3.26)**	**<0.001**
Hepatic insufficiency (%)	206 (7.45)	72 (7.94)	0.810		
ALT, U/L	27.60 ± 115.65	30.10 ± 80.68	**<** **0.001**		
NYHA II~IV (%)	33 (1.17)	88 (9.66)	**<0.001**	**9.03 (6.01~13.57)**	**<0.001**
NT-proBNP, pg/mL	2420.61 ± 7454.93	3883.61 ± 7504.02	**<0.001**		
Stroke (%)	424 (15.32)	193 (21.28)	**<** **0.001**		

Note: The abbreviations are shown above. The bold value indicates statistically significant difference.

**Table 3 biomedicines-13-01557-t003:** Baseline characteristics of cohort: recurrence in diabetic patients with AF after RFA at Tongji Hospital.

Variables	Recurrence (n = 9)	Non-Recurrence (n = 30)	*p*
Age, years	62.89 ± 11.10	67.97 ± 8.04	0.317
Male (%)	7 (77.78)	23 (76.67)	1.000
Persistent atrial fibrillation (%)	5 (55.56)	6 (20.00)	0.085
Hypertension (%)	7 (77.78)	17 (56.67)	0.437
Coronary heart disease (%)	1 (11.11)	13 (43.33)	0.119
CKD (%)	4 (44.44)	4 (13.33)	0.065
Pulmonary infection (%)	0 (0.00)	2 (6.67)	1.000
Chronic bronchitis (%)	0 (0.00)	2 (6.67)	1.000
Hyperthyroidism (%)	0	0	-
NYHA II~IV (%)	4 (44.44)	4 (13.33)	0.065
LAD, mm	4.71 ± 0.60	4.38 ± 0.59	0.141

Note: CKD, chronic kidney disease; NYHA II~IV, New York Heart Association II~IV; LAD, diameter of the left atrium; AF, atrial fibrillation; RFA, radiofrequency ablation.

**Table 4 biomedicines-13-01557-t004:** Baseline data before and after excluding confounding subjects in metabolomics analysis.

Variables	Before		*p*	After		*p*
	Case (n = 10)	Control (n = 10)		Case (n = 7)	Control (n = 6)	
Age, years	56.60 ± 10.13	62.90 ± 6.79	0.101	57.29 ± 12.34	63.50 ± 7.29	0.351
Male, (%)	9 (90.00)	3 (30.00)	0.020	6 (85.71)	3 (50.00)	0.070
NYHA II~IV, (%)	5 (50.00)	0 (0.00)	0.033	5 (71.43)	0 (0.00)	0.021
NT-BNP, pg/mL	1474.23 ± 1190.46	48.29 ± 45.09	<0.001	1480.76 ± 1457.95	62.48 ± 53.10	0.003
EF, %	51.10 ± 7.43	63.50 ± 5.13	0.003	53.29 ± 8.01	63.33 ± 4.84	0.022
E, cm/s	119.50 ± 32.86	70.56 ± 13.97	0.002	125.71 ± 38.33	66.20 ± 13.88	0.008
E′, cm/s	7.10 ± 1.10	5.57 ± 0.79	0.005	7.57 ± 0.98	5.40 ± 0.89	0.003
E/E′	17.17 ± 5.14	11.89 ± 1.67	0.034	17.02 ± 6.29	12.31 ± 1.84	0.139
LAD, mm	4.99 ± 0.88	3.77 ± 0.72	0.005	5.50 ± 0.39	3.78 ± 0.69	<0.001
LVEDD, mm	5.08 ± 0.53	4.42 ± 0.76	0.155	5.29 ± 0.51	4.70 ± 0.13	0.022
Hypertension, (%)	8 (80.00)	7 (70.00)	1.000	5 (71.43)	5 (83.33)	1.000
CKD, (%)	8 (80.00)	1 (10.00)	0.005	5 (71.43)	1 (16.67)	0.103
CHD, (%)	5 (50.00)	8 (80.00)	0.350	5 (71.43)	4 (66.67)	0.286

Note: NYHA II~IV, New York Heart Association II~IV; NT-BNP, N-terminal pro-B-type natriuretic peptide; EF, ejection fraction; LAD, diameter of the left atrium; LVEDD, left ventricular end-diastolic diameter; CKD, chronic kidney disease; CHD, coronary heart disease.

## Data Availability

The data that support the findings of this study are available on request from the corresponding author upon reasonable request.

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
