# Peer review of "Identifying High-Risk Atrial Fibrillation in Diabetes: Evidence from Nomogram and Plasma Metabolomics Analysis"

_biomedicines, 2025, doi:10.3390/biomedicines13071557_

Round 1
Reviewer 1 Report
Comments and Suggestions for Authors
The manuscript by Q. Luo et al. describes a model proposed by the authors to identify patients with diabetes at high risk of atrial fibrillation (AF). This model was developed by comparing cohorts of patients with both diabetes and AF (the case group) with patients with diabetes only (the control group). These patient groups were formed using data from the National Health and Nutrition Examination Survey (NHANES) and were validated using patient data from Tongji Hospital (China), which was used as the test dataset.
Logistic regression analysis, followed by ROC analysis, revealed that the model sufficiently reliably assesses AF in diabetic populations (AFDB score).
An additional untargeted metabolomic analysis revealed some differences in metabolites in the case group compared to the control group.
Overall, the work gives the impression of a comprehensive clinical and metabolomic study.
Comments:
Section 3.5: Some conclusions are necessary to explain the sufficiency or insufficiency of the obtained data.
Section 3.6 (lines 217–236): The authors characterise the data presented in Figures 5–7 and point to a number of metabolic pathway abnormalities found in the case group. What is the possible relationship between these abnormalities and the development of AF in diabetes?
Do the identified factors relate to both type 1 and type 2 diabetes, or only one of them?
Author Response
Reviewer 1:
Thank you for your advice! Your advice has given us great inspiration. Next are the point-to-point responses.
Comments:
Comment 1:
Section 3.5: Some conclusions are necessary to explain the sufficiency or insufficiency of the obtained data.
Response 1:
Lines 216-236: Figure 4 demonstrated the evaluation for the AFDB score for the NHANES and the Tongji Hospital. Figure 4a displayed the ROC curve for the AFDB score based on the NHANES database, with an AUC of 0.85. Figure 4b, 4c, 4d, and 4e represented the calibration curve, DCA curve, confusion matrix, and CIC curve of the train dataset, respectively. The DCA curve indicated that the AFDB score provided good clinical net benefit in the training set. The calibration curve closely aligned with the ideal reference line, suggesting that the model has high predictive accuracy. Figure 4f showed the KM curve for survival prognosis based on the AFDB score for the NHANES dataset, distinguishing high-risk and low-risk populations based on the median score (The median score was 90.72.). It could be observed that the high-risk population had significantly worse survival prognosis compared to the low-risk group (P < 0.001). Figure 4g showed the ROC curve for the AFDB score based on the data from Tongji Hospital, with an AUC of 0.78. Figure 4h, 4i, 4j, and 4k represented the calibration curve, DCA curve, confusion matrix, and CIC curve of the test dataset, respectively. The DCA curve indicated that the AFDB score provided a certain level of clinical net benefit in the testing set. The calibration curve’s alignment with the ideal reference line suggests that the model has a certain degree of predictive accuracy. However, compared to the results in the training set, the model performed less favorably, indicating that its generalizability in external testing still has room for improvement. Figure 4l showed the AFDB score's predictive value for post-RFA recurrence in the Tongji hospital follow-up cohort, with the combination of left atrial diameter and preoperative AF type improving the prediction accuracy to 0.72.
Comment 2:
Section 3.6 (lines 217–236): The authors characterize the data presented in Figures 5–7 and point to a number of metabolic pathway abnormalities found in the case group. What is the possible relationship between these abnormalities and the development of AF in diabetes?
Do the identified factors relate to both type 1 and type 2 diabetes, or only one of them?
Response 2:
Lines 544-613: It should be emphasized that both diabetes and AF are highly heterogeneous diseases. Diabetes not only includes type 1 and type 2 diabetes, but also broadly encompasses gestational diabetes, mitochondrial diabetes, and diabetes secondary to exocrine pancreatic disorders[1]. Similarly, AF can be classified based on episode characteristics and duration into paroxysmal, persistent, and permanent AF, and can also be divided etiologically into valvular and non-valvular AF[2]. Given the significant variability in individual pathophysiological mechanisms, elucidating the disease mechanisms on a personalized basis is crucial for understanding the onset and progression of these conditions.
Wang et al. reviewed the specific mechanisms underlying the increased incidence of AF in patients with diabetes. On one hand, these mechanisms include glycemic variability, inflammatory responses, and oxidative stress; on the other hand, they involve structural remodeling of the heart, electromechanical remodeling, electrical remodeling, and autonomic remodeling[3]. Firstly, both the arachidonic acid metabolism pathway and its upstream linoleic acid metabolism pathway, which were identified in the enrichment analysis, are involved in inflammatory responses[4]. In addition, the elevated plasma levels of cortisol served as evidence of stress in the case group. Moreover, increased levels of various steroid hormones other than cortisol also suggested adrenal cortex hyperfunction, reflecting both stress and autonomic remodeling[5]. Furthermore, as the heart is a highly energy-demanding organ, diabetic patients often experience impaired cardiac energy metabolism. Hahn et al. conducted a metabolomic analysis using plasma and myocardial samples from patients with heart failure with preserved ejection fraction (HFpEF). The metabolic profile they summarized— “HFpEF is a syndrome with substantial fuel inflexibility”—is highly consistent with the findings in our metabolomic analysis[6]. Notably, both AF and diabetes are major comorbidities of HFpEF[7,8]. Thus, this substantial fuel inflexibility also exists in high-risk AF patients with diabetes, characterized by the accumulation of long-chain acylcarnitines, increased branched-chain amino acids, buildup of tricarboxylic acid cycle intermediates, disrupted ATP metabolic homeostasis, and overall mitochondrial dysfunction.
Therefore, the metabolic characteristics of AF in diabetes summarized in this study, which referred to more significant disruption of energy utilization, inflammation, and stress responses, align with the underlying pathophysiology of both diabetes and AF observed in previous studies. These molecular-level mechanistic features are often present across different clinical phenotypes of diabetes and AF. Meanwhile, numerous studies summarized the characteristics of metabolomics, proteomics, genomics, and transcriptomics in AF or diabetes.[9-11]Integrating multi-omics data (with a focus on plasma metabolomics in this study) greatly aids our understanding of the mechanisms underlying high-risk atrial fibrillation in diabetes.
Precisely because of the heterogeneity of diabetes and AF, this study did not use "black-box" artificial intelligence algorithms in the modeling process of the Electronic Health Record-integrated (EHR-integrated) too[12,13]. Instead, variable selection was based on traditional p-values and OR, and modeling was performed using logistic regression. Additionally, results from plasma metabolomics of the case and control groups from our center were incorporated to help explain potential pathogenic mechanisms. The reason for this approach lies in fully considering the target users of the tool—clinicians. In practical applications, clinical prediction models often face issues of overfitting and underfitting[14]. Overfitting occurs when the model fits the training data too closely but has poor generalizability. Compared with artificial intelligence models, ours is more transparent, so if overfitting occurs, users (especially clinicians) can interpret and adjust the AFDB nomogram based on pathophysiological knowledge, characteristics of comorbidities in their own patients, and plasma metabolomic profiles. Underfitting occurs when the model does not fit the training data adequately and fails to capture underlying patterns. Since the modeling in this study used NHANES, a publicly available and transparent database, if underfitting arises, users (especially clinicians) can expand the AFDB nomogram training set using their own local data or multi-omics data (such as the metabolomics data in this study) to improve the model’s performance across various scenarios.
On one hand, the advantage of the EHR-integrated tool is that it is derived from clinical data, making it relatively easy to obtain and apply. However, its limitation lies in the presence of bias when applied to different populations. In the NHANES dataset in this study, the variable with the highest OR was chronic bronchitis; while in the validation data from Tongji Hospital, the highest OR was for NYHA II–IV. This discrepancy is due to admission rate bias, which also affected the performance of the AFDB score in our hospital. On the other hand, plasma metabolomics offers a deeper mechanistic understanding, facilitating individualized interpretation of disease characteristics in different populations. Its drawbacks are that it is not routinely performed in clinical laboratories, has relatively high costs, and it is difficult for a single center to obtain large-sample data.
Yao et al. demonstrated that combining clinical and polygenic risk scores yielded the highest predictive accuracy for AF[15]. Therefore, the comprehensive integration and interpretation of different types of data will promote personalized management and precision treatment of diabetes with AF.
Reference:
- Seino, Y.; Nanjo, K.; Tajima, N.; Kadowaki, T.; Kashiwagi, A.; Araki, E.; Ito, C.; Inagaki, N.; Iwamoto, Y.; Kasuga, M.; et al. Report of the committee on the classification and diagnostic criteria of diabetes mellitus. J Diabetes Investig 2010, 1, 212-228, doi:10.1111/j.2040-1124.2010.00074.x.
- Joglar, J.A.; Chung, M.K.; Armbruster, A.L.; Benjamin, E.J.; Chyou, J.Y.; Cronin, E.M.; Deswal, A.; Eckhardt, L.L.; Goldberger, Z.D.; Gopinathannair, R.; et al. 2023 ACC/AHA/ACCP/HRS Guideline for the Diagnosis and Management of Atrial Fibrillation: A Report of the American College of Cardiology/American Heart Association Joint Committee on Clinical Practice Guidelines. J Am Coll Cardiol 2024, 83, 109-279, doi:10.1016/j.jacc.2023.08.017.
- Wang, A.; Green, J.B.; Halperin, J.L.; Piccini, J.P., Sr. Atrial Fibrillation and Diabetes Mellitus: JACC Review Topic of the Week. J Am Coll Cardiol 2019, 74, 1107-1115, doi:10.1016/j.jacc.2019.07.020.
- Wang, B.; Wu, L.; Chen, J.; Dong, L.; Chen, C.; Wen, Z.; Hu, J.; Fleming, I.; Wang, D.W. Metabolism pathways of arachidonic acids: mechanisms and potential therapeutic targets. Signal Transduct Target Ther 2021, 6, 94, doi:10.1038/s41392-020-00443-w.
- Golub, M.S. The adrenal and the metabolic syndrome. Curr Hypertens Rep 2001, 3, 117-120, doi:10.1007/s11906-001-0022-x.
- Hahn, V.S.; Petucci, C.; Kim, M.S.; Bedi, K.C., Jr.; Wang, H.; Mishra, S.; Koleini, N.; Yoo, E.J.; Margulies, K.B.; Arany, Z.; et al. Myocardial Metabolomics of Human Heart Failure With Preserved Ejection Fraction. Circulation 2023, 147, 1147-1161, doi:10.1161/circulationaha.122.061846.
- Park, J.J. Epidemiology, Pathophysiology, Diagnosis and Treatment of Heart Failure in Diabetes. Diabetes Metab J 2021, 45, 146-157, doi:10.4093/dmj.2020.0282.
- Kotecha, D.; Lam, C.S.; Van Veldhuisen, D.J.; Van Gelder, I.C.; Voors, A.A.; Rienstra, M. Heart Failure With Preserved Ejection Fraction and Atrial Fibrillation: Vicious Twins. J Am Coll Cardiol 2016, 68, 2217-2228, doi:10.1016/j.jacc.2016.08.048.
- Linna-Kuosmanen, S.; Vuori, M.; Kiviniemi, T.; Palmu, J.; Niiranen, T. Genetics, transcriptomics, metagenomics, and metabolomics in the pathogenesis and prediction of atrial fibrillation. Eur Heart J Suppl 2024, 26, iv33-iv40, doi:10.1093/eurheartjsupp/suae072.
- Jin, Q.; Ma, R.C.W. Metabolomics in Diabetes and Diabetic Complications: Insights from Epidemiological Studies. Cells 2021, 10, doi:10.3390/cells10112832.
- Jia, W.; Chan, J.C.; Wong, T.Y.; Fisher, E.B. Diabetes in China: epidemiology, pathophysiology and multi-omics. Nat Metab 2025, 7, 16-34, doi:10.1038/s42255-024-01190-w.
- Zhang, Y.; Li, S.; Mai, P.; Yang, Y.; Luo, N.; Tong, C.; Zeng, K.; Zhang, K. A machine learning-based model for predicting paroxysmal and persistent atrial fibrillation based on EHR. BMC Med Inform Decis Mak 2025, 25, 51, doi:10.1186/s12911-025-02880-5.
- Zhou, Y.; Zhang, D.; Chen, Y.; Geng, S.; Wei, G.; Tian, Y.; Shi, L.; Wang, Y.; Hong, S.; Liu, X. Screening Tool for Paroxysmal Atrial Fibrillation Based on a Deep-Learning Algorithm Using Printed 12-Lead Electrocardiographic Records during Sinus Rhythm. Rev Cardiovasc Med 2024, 25, 242, doi:10.31083/j.rcm2507242.
- Wagner, M.W.; Namdar, K.; Biswas, A.; Monah, S.; Khalvati, F.; Ertl-Wagner, B.B. Radiomics, machine learning, and artificial intelligence-what the neuroradiologist needs to know. Neuroradiology 2021, 63, 1957-1967, doi:10.1007/s00234-021-02813-9.
- Yao, Y.; Zhang, M.J.; Wang, W.; Zhuang, Z.; He, R.; Ji, Y.; Knutson, K.A.; Norby, F.L.; Alonso, A.; Soliman, E.Z.; et al. Multimodal data integration to predict atrial fibrillation. Eur Heart J Digit Health 2025, 6, 126-136, doi:10.1093/ehjdh/ztae081.

Reviewer 2 Report
Comments and Suggestions for Authors
Manuscript ID: biomedicines-3689284
This study developed and validated a clinical nomogram and performed plasma metabolomics to identify high-risk atrial fibrillation (AF) in diabetic patients. Both methods showed strong predictive value, and their combined use highlights the potential for improved AF risk stratification and personalized management in diabetes. The manuscript may be further improved by following suggestions.
- The word already mentioned in the title should not be repeated in keyword, select correct keyword relevant to the study.
- Use effect size for all statistical analyses to quantify the magnitude of differences or relationships, providing more meaningful insights beyond mere statistical significance.
- The abstract contains overly technical metrics (e.g., AUC values, exact sample sizes). Consider simplifying the language for better readability and emphasizing key findings and their implications.
- The metabolomics sample size (n=7 AF cases vs. n=6 controls) is very limited. Please justify how this sample size provides adequate statistical power to support metabolomic conclusions. Consider discussing this limitation more explicitly in the discussion.
- Although both a clinical nomogram and metabolomics analysis are presented, the integration between these two approaches is minimal. A comparative or correlational analysis linking metabolite profiles with nomogram scores could strengthen the case for combined risk prediction.
- The process of variable selection for the final multivariable logistic model needs more transparency. Provide criteria for exclusion/inclusion (e.g., multicollinearity, VIF thresholds) and consider including a supplementary table listing full regression outputs.
- The test cohort from Tongji Hospital is only briefly described. Provide more information on patient demographics, comorbidities, and clinical characteristics. Were there any differences between the train and test sets?
- The discussion sometimes becomes repetitive (e.g., frequent reiteration of the nomogram’s AUC). Try to condense overlapping content and emphasize broader implications and future directions instead.
- Several figures (e.g., Figures 5-7) lack sufficient explanation in their captions. For each figure, briefly state what the key finding is and how it supports the conclusions of the study.
- The manuscript would benefit from a section discussing how the AFDB nomogram could be practically used in clinical settings e.g., as an EHR-integrated tool and what barriers to adoption might exist.
May be improved.
Author Response
Reviewer 2:
Thank you for your advice! Your advice has given us great inspiration. Next are the point-to-point responses.
Comments:
Comment 1:
The word already mentioned in the title should not be repeated in keyword, select correct keyword relevant to the study.
Response 1:
Keywords: NHANES; Liquid chromatography-mass spectrometry; Integrated strategies; Personalized management
Comment 2:
Use effect size for all statistical analyses to quantify the magnitude of differences or relationships, providing more meaningful insights beyond mere statistical significance.
Response 2:
Lines 600-606: the advantage of the EHR-integrated tool is that it is derived from clinical data, making it relatively easy to obtain and apply. However, its limitation lies in the presence of bias when applied to different populations. In the NHANES dataset in this study, the variable with the highest OR was chronic bronchitis; while in the validation data from Tongji Hospital, the highest OR was for NYHA II–IV. This discrepancy is due to admission rate bias, which also affected the performance of the AFDB score in our hospital.
Comment 3:
The abstract contains overly technical metrics (e.g., AUC values, exact sample sizes). Consider simplifying the language for better readability and emphasizing key findings and their implications.
Response 3:
Abstract: Background: Diabetes significantly increases the risk of atrial fibrillation (AF), but identifying high-risk individuals remains a clinical challenge. This study aimed to improve AF risk stratification in diabetic patients through a combination of clinical modeling and untargeted metabolomic analysis. Methods: A clinical risk score was developed using data from the National Health and Nutrition Examination Survey (NHANES) and validated in an independent cohort from Tongji Hospital. Its association with long-term outcomes and its ability to predict AF recurrence after catheter ablation were assessed in follow-up studies. Additionally, untargeted plasma metabolomics was performed in a subset of diabetic patients with and without AF to explore underlying mechanism. Results: The risk score showed good predictive performance in both the development and validation cohorts and was significantly associated with clinical prognosis. When combined with left atrial diameter and AF type, it also improved the prediction of AF recurrence after ablation. Metabolomic profiling revealed notable disturbances in energy metabolism, heightened inflammatory activity, and elevated stress responses in AF patients, indicating a distinct metabolic risk profile. Conclusions: This study provided two approaches to identify high-risk AF in diabetic patients, discussed the underlying pathophysiological mechanisms, and compared their characteristics and applications. And integrated strategies could improve AF risk stratification and personalized management in the diabetic.
Comment 4:
The metabolomics sample size (n=7 AF cases vs. n=6 controls) is very limited. Please justify how this sample size provides adequate statistical power to support metabolomic conclusions. Consider discussing this limitation more explicitly in the discussion.
Response 4:
Lines 614-629: It is worth noting that the sample size and its representativeness often influence the reliability of conclusions. First, Gumprecht et al. conducted a prospective study and found that the incidence of AF among diabetic patients in the Polish population was approximately 25%. The AFDB score includes 10 independent variables, and based on the Events Per Variable (EPV) method for sample size estimation (with EPV set at 10 and the training set proportion at 0.52), at least 400 cases are needed in the training set[1]. Considering a 10% rate of invalid samples, a total sample size of at least 855 cases is required. The sample size used in this part of our study fully meets this requirement. In addition, we used G*Power 3.1 (http://www.gpower.hhu.de) to calculate the classification power of the plasma metabolites shown in Figure 7 (n = 7 AF cases vs. n = 6 controls)[2]. The statistical method used was a two-tailed t-test for two independent groups with a significance level of P = 0.05. The minimum classification powers for the three representative metabolites involved in the mechanisms shown in Figures 7b, 7e, and 7f—namely isoleucine, TXB2, and cortisone—were 1.00, 1.00, and 0.91, respectively, all exceeding the threshold of 0.80. Appendix C shows the means and standard deviations of these three metabolites4 in the two groups. Future studies require larger, multicenter cohorts.
Appendix C
Table C1. The means and standard deviations of the 3 representative metabolites in the 2 groups.
|
Variables |
||||
|
Isoleucine |
1665.08 |
247.48 |
185.30 |
14.15 |
|
TXB2 |
1320.34 |
131.70 |
569.70 |
125.73 |
|
Cortisone |
9094.37 |
3132.71 |
2725.98 |
2564.10 |
Note: The case group is group 1 (n=7), and the control group is group 2 (n=6). The abbreviations are provided in the main text.
Comment 5:
Although both a clinical nomogram and metabolomics analysis are presented, the integration between these two approaches is minimal. A comparative or correlational analysis linking metabolite profiles with nomogram scores could strengthen the case for combined risk prediction.
Response 5:
Lines 600-613: On one hand, the advantage of the EHR-integrated tool is that it is derived from clinical data, making it relatively easy to obtain and apply. However, its limitation lies in the presence of bias when applied to different populations. In the NHANES dataset in this study, the variable with the highest OR was chronic bronchitis; while in the validation data from Tongji Hospital, the highest OR was for NYHA II–IV. This discrepancy is due to admission rate bias, which also affected the performance of the AFDB score in our hospital. On the other hand, plasma metabolomics offers a deeper mechanistic understanding, facilitating individualized interpretation of disease characteristics in different populations. Its drawbacks are that it is not routinely performed in clinical laboratories, has relatively high costs, and it is difficult for a single center to obtain large-sample data.
Yao et al. demonstrated that combining clinical and polygenic risk scores yielded the highest predictive accuracy for AF[3]. Therefore, the comprehensive integration and interpretation of different types of data will promote personalized management and precision treatment of diabetes with AF.
Comment 6:
The process of variable selection for the final multivariable logistic model needs more transparency. Provide criteria for exclusion/inclusion (e.g., multicollinearity, VIF thresholds) and consider including a supplementary table listing full regression outputs.
Response 6:
Line 128-129: Collinearity analysis is provided in Appendix B, with a VIF > 10 considered suggestive of strong collinearity and taken into account as a reference during variable assessment.
Line 207-209: Collinearity analysis shown in Appendix B served as a reference, indicating that leukocytes, neutrophils, and lymphocytes exhibited notable collinearity in the univariate logistic regression.
Appendix B
Table B1. Collinearity analysis in univariate logistic regression.
|
Variables |
VIF |
Tolerance |
|
Age |
1.320 |
0.757 |
|
Male |
1.326 |
0.754 |
|
Non-Hispanic white |
1.207 |
0.829 |
|
BMI |
1.158 |
0.864 |
|
Smoke |
1.114 |
0.898 |
|
HbA1c |
1.458 |
0.686 |
|
Hypertension |
1.134 |
0.882 |
|
Systolic pressure |
1.278 |
0.782 |
|
Diastolic pressure |
1.251 |
0.799 |
|
Infection |
1.830 |
0.547 |
|
Leukocytes |
56.236 |
0.018 |
|
Neutrophils |
39.020 |
0.026 |
|
Lymphocytes |
13.318 |
0.075 |
|
Platelets |
1.297 |
0.771 |
|
CRP |
1.287 |
0.777 |
|
Chronic bronchitis |
1.290 |
0.775 |
|
Hyperlipidemia |
2.075 |
0.482 |
|
Triglycerides |
1.209 |
0.827 |
|
LDL-C |
1.362 |
0.734 |
|
HDL-C |
1.224 |
0.817 |
|
TC |
2.436 |
0.411 |
|
CKD |
1.830 |
0.546 |
|
CreatinineL |
1.927 |
0.519 |
|
Anemia |
1.164 |
0.859 |
|
Hemoglobin |
1.556 |
0.643 |
|
Hyperthyroidism |
1.977 |
0.506 |
|
FT4 |
2.131 |
0.469 |
|
TSH |
1.130 |
0.885 |
|
Coronary heart disease |
1.213 |
0.825 |
|
Hepatic insufficiency |
2.145 |
0.466 |
|
ALT |
2.213 |
0.452 |
|
NYHA II~IV |
1.399 |
0.715 |
|
NT-proBNP |
1.541 |
0.649 |
Note: The abbreviations are shown in the main text.
Table B2. Collinearity analysis in the preliminary multivariate logistic regression.
|
Variables |
VIF |
Tolerance |
|
Age |
1.154 |
0.867 |
|
Male |
1.118 |
0.895 |
|
Non-Hispanic white |
1.090 |
0.917 |
|
Smoke |
1.090 |
0.918 |
|
Hypertension |
1.076 |
0.929 |
|
Infection |
1.037 |
0.964 |
|
Chronic bronchitis |
1.266 |
0.79 |
|
CKD |
1.060 |
0.943 |
|
Hyperthyroidism |
1.014 |
0.986 |
|
Coronary heart disease |
1.191 |
0.839 |
|
NYHA II~IV |
1.372 |
0.729 |
Note: The abbreviations are shown in the main text.
Table B3. Collinearity analysis in the final multivariate logistic regression.
|
Variables |
VIF |
Tolerance |
|
Age |
1.147 |
0.872 |
|
Male |
1.046 |
0.956 |
|
Non-Hispanic white |
1.089 |
0.918 |
|
Hypertension |
1.075 |
0.93 |
|
Infection |
1.032 |
0.969 |
|
Chronic bronchitis |
1.258 |
0.795 |
|
CKD |
1.060 |
0.943 |
|
Hyperthyroidism |
1.014 |
0.986 |
|
Coronary heart disease |
1.191 |
0.839 |
|
NYHA II~IV |
1.371 |
0.730 |
Note: The abbreviations are shown in the main text.
Comment 7:
The test cohort from Tongji Hospital is only briefly described. Provide more information on patient demographics, comorbidities, and clinical characteristics. Were there any differences between the train and test sets?
Response 7:
Line-602-606: In the NHANES dataset in this study, the variable with the highest OR was chronic bronchitis; while in the validation data from Tongji Hospital, the highest OR was for NYHA II–IV. This discrepancy is due to admission rate bias, which also affected the performance of the AFDB score in our hospital
Table 2. Baseline characteristics at Tongji Hospital and univariate regression analysis of specific variables.
|
Variables |
Control (n = 2819) |
Case (n = 911) |
P |
OR (95%CI) |
P(OR) |
|
Age, years |
57.85 ± 15.31 |
71.79 ± 9.93 |
<0.001 |
1.09 (1.08 ~ 1.10) |
<0.001 |
|
Male (%) |
1557 (55.23) |
556 (61.03) |
0.002 |
1.27 (1.09 ~ 1.48) |
0.002 |
|
Han Chinese (%) |
2803 (99.43) |
907 (99.56) |
0.841 |
0.77 (0.26 ~ 2.32) |
0.645 |
|
BMI, kg/m^2 |
23.66 ± 12.47 |
24.87 ± 4.44 |
<0.001 |
|
|
|
Smoke (%) |
252 (9.12) |
122 (13.45) |
<0.001 |
|
|
|
HbA1c, % |
8.40 ± 5.04 |
7.50 ± 1.58 |
<0.001 |
|
|
|
Hypertension (%) |
1302 (46.19) |
554 (60.81) |
<0.001 |
1.81 (1.55 ~ 2.11) |
<0.001 |
|
Systolic pressure, mmHg |
127.86 ± 41.48 |
137.35 ± 344.38 |
0.410 |
|
|
|
Diastolic pressure, mmHg |
78.35 ± 15.78 |
103.23 ± 29.82 |
<0.001 |
|
|
|
Pulmonary infection (%) |
331 (11.74) |
211 (23.16) |
<0.001 |
2.27 (1.87 ~ 2.75) |
<0.001 |
|
Leukocytes,1000 cells/uL |
7.26 ± 3.21 |
7.45 ± 4.56 |
0.295 |
|
|
|
Neutrophils,1000 cells/uL |
5.02 ± 2.99 |
5.32 ± 4.19 |
0.753 |
|
|
|
Lymphocytes,1000 cells/uL |
1.57 ± 0.72 |
1.40 ± 1.06 |
<0.001 |
|
|
|
Platelets, 1000 cells/uL |
226.32 ± 415.24 |
195.05 ± 80.46 |
<0.001 |
|
|
|
CRP, mg/L |
18.98 ± 41.61 |
24.79 ± 51.03 |
0.028 |
|
|
|
Chronic bronchitis (%) |
49 (1.74) |
45 (4.94) |
<0.001 |
2.94 (1.95 ~ 4.43) |
<0.001 |
|
Hyperlipidemia (%) |
723 (26.21) |
155 (17.09) |
<0.001 |
|
|
|
Triglycerides, mmol/L |
2.92 ± 11.46 |
1.71 ± 1.22 |
<0.001 |
|
|
|
LDL-C, mmol/L |
2.80 ± 9.23 |
2.08 ± 0.84 |
<0.001 |
|
|
|
HDL-C, mmol/L |
1.10 ± 0.45 |
0.98 ± 0.30 |
<0.001 |
|
|
|
TC, mmol/L |
4.68 ± 10.85 |
3.66 ± 1.06 |
<0.001 |
|
|
|
CKD (%) |
551 (19.55) |
344 (37.76) |
<0.001 |
2.50 (2.12 ~ 2.94) |
<0.001 |
|
Creatinine, µmol/L |
115.38 ± 153.62 |
124.51 ± 140.24 |
0.122 |
|
|
|
Anemia (%) |
109 (12.02) |
422 (15.27) |
0.008 |
|
|
|
Hemoglobin, g/L |
130.85 ± 231.79 |
123.64 ± 26.77 |
0.489 |
|
|
|
Hyperthyroidism (%) |
29 (1.03) |
23 (2.52) |
<0.001 |
2.49 (1.43 ~ 4.33) |
0.001 |
|
FT4, pmol/L |
14.92 ± 35.40 |
16.79 ± 8.89 |
<0.001 |
|
|
|
TSH, uIU/mL |
2.36 ± 3.99 |
3.09 ± 3.96 |
0.009 |
|
|
|
Coronary heart disease (%) |
558 (19.79) |
370 (40.61) |
<0.001 |
2.77 (2.36 ~ 3.26) |
<0.001 |
|
Hepatic insufficiency (%) |
206 (7.45) |
72 (7.94) |
0.810 |
|
|
|
ALT, U/L |
27.60 ± 115.65 |
30.10 ± 80.68 |
<0.001 |
|
|
|
NYHA II ~IV (%) |
33 (1.17) |
88 (9.66) |
<0.001 |
9.03 (6.01 ~ 13.57) |
<0.001 |
|
NT-proBNP, pg/ml |
2420.61 ± 7454.93 |
3883.61 ± 7504.02 |
<0.001 |
|
|
|
Stroke (%) |
424 (15.32) |
193 (21.28) |
<0.001 |
|
|
Note: The abbreviations are shown above. The bold value indicates statistically significant difference.
Comment 8:
The discussion sometimes becomes repetitive (e.g., frequent reiteration of the nomogram’s AUC). Try to condense overlapping content and emphasize broader implications and future directions instead.
Response 8:
Line-512-543: Numerous studies have developed risk scores for predicting AF risk. Meanwhile, Schnabel et al., Chamberlai et al., FIND-AF model, The CHARGE-AF score identified a lot of clinical factors associated with AF occurrence[4-7]. However, these models had several limitations, including reliance on single-center or single-source data, absence of cross-ethnic external validation, lack of comparative analysis between nomogram and metabolomics approaches, dependence on complex algorithms with limited interpretability, and restricted generalizability. Notably, none of these models were specifically developed for diabetic populations. It was noteworthy that the C2HEST score, similar to our model in terms of simplicity and interpretability, which performed worse than ours[8]. This performance gap might be attributed to the C2HEST score's lack of specificity for diabetic populations. This performance gap might be attributed to the C2HEST score's lack of specificity for diabetic populations. Meanwhile, our scoring system drew on diverse datasets from North America to Asia, including both U.S. and Chinese populations, which complemented each other and demonstrated a broad application range and strong generalizability.
The latest guideline not only emphasized the importance of preventing AF before its electrocardiographic confirmation by introducing the concepts of AF stage 1 and stage 2, but also highlighted that, in addition to rate control and restoration of sinus rhythm, managing risk factors and addressing atrial structural remodeling were crucial for the prevention and treatment of AF[9]. The nomogram in this study included variables such as age, sex, ethnicity, hypertension, CKD, coronary heart disease, and others. While factors like age and sex are non-modifiable, the remaining conditions are treatable. Hypertension, CKD, heart failure and coronary heart disease are common diabetes complications. In other words, lowering blood pressure, improving heart failure, treating coronary heart disease, controlling infections (or pulmonary infection), improving renal function, controlling COPD (or chronic bronchitis), and managing hyperthyroidism were crucial for the prevention of AF. Furthermore, diabetic subjects with higher AFDB scores (high-risk diabetic population for AF) had a higher risk of mortality (P < 0.01) and shorter follow-up time (P < 0.01). Therefore, preventing and managing AF in diabetic patients was of great importance for improving survival and prognosis, which suggested related measures could improve prognosis and prolong survival in the diabetic. In addition, this score also provided certain guidance for predicting the recurrence of AF after surgery.
Line544-613: This section incorporates comments from both reviewers and provides a comparative discussion summarizing the relationship between plasma metabolomics and underlying mechanisms, as well as the respective characteristics of the two approaches.
Comment 9:
Several figures (e.g., Figures 5-7) lack sufficient explanation in their captions. For each figure, briefly state what the key finding is and how it supports the conclusions of the study.
Response 9:
Line 443-450:
Figure 5. Overview of the metabolisms difference of the 2 groups before excluding confounding subjects.
(a)PCA between 2 groups before excluding confounding subjects. (b)OP-LS before excluding confounding subjects. (c) Volcano plot before excluding confounding subjects. (d) Pie chart before excluding confounding subjects. (e) Heatmap before excluding confounding subjects. (f) KEGG enrichment analysis before excluding confounding subjects. Note: AF: atrial fibrillation; PCA, principal component analysis; OP-LS, orthogonal partial least squares; KEGG, Kyoto Encyclopedia of Genes and Genomes.
Line452-458:
Figure 6. Overview of the metabolisms difference of the 2 groups after excluding confounding subjects.
(a)PCA after excluding confounding subjects. (b)OP-LS after excluding confounding subjects. (c) Volcano plot after excluding confounding subjects. (d) Pie chart after excluding confounding subjects. (e) Heatmap after excluding confounding subjects. (f) KEGG enrichment analysis after excluding confounding subjects. Note: AF: atrial fibrillation; PCA, principal component analysis; OP-LS, orthogonal partial least squares; KEGG, Kyoto Encyclopedia of Genes and Genomes.
Line 460-466
Figure 7. The plasma metabolic characteristics of high-risk AF in diabetes.
(a) Radar plot of energy metabolism disruption. (b) Differences in the relative content of isoleucine and ROC curve. (c) Radar plot of inflammatory response. (d) Differences in the relative content of TXB2 and ROC curve. (e) Radar plot of corticosteroids, indicating a stronger stress response. (f) Differences in the relative content of cortisol and ROC curve. Note: AC, acylcarnitine; TXB2, thromboxane B2; HETE, Hydroxyeicosatetraenoic acid; LTB4, leukotriene B4. The data in the radar chart were log10-transformed. **p < 0.01.
Line544-613: This section incorporates comments from both reviewers and provides a comparative discussion summarizing the relationship between plasma metabolomics and underlying mechanisms, as well as the respective characteristics of the two approaches. And this part also stated what the key finding is and how it supports the conclusions of the study
Comment 10:
The manuscript would benefit from a section discussing how the AFDB nomogram could be practically used in clinical settings e.g., as an EHR-integrated tool and what barriers to adoption might exist.
Response 10:
Line-588-599: In practical applications, clinical prediction models often face issues of overfitting and underfitting[10]. Overfitting occurs when the model fits the training data too closely but has poor generalizability. Compared with artificial intelligence models, ours is more transparent, so if overfitting occurs, users (especially clinicians) can interpret and adjust the AFDB nomogram based on pathophysiological knowledge, characteristics of comorbidities in their own patients, and plasma metabolomic profiles. Underfitting occurs when the model does not fit the training data adequately and fails to capture underlying patterns. Since the modeling in this study used NHANES, a publicly available and transparent database, if underfitting arises, users (especially clinicians) can expand the AFDB nomogram training set using their own local data or multi-omics data (such as the metabolomics data in this study) to improve the model’s performance across various scenarios.
Reference:
- Gumprecht, J.; Lip, G.Y.H.; Sokal, A.; Średniawa, B.; Mitręga, K.; Stokwiszewski, J.; Wierucki, Ł.; Rajca, A.; Rutkowski, M.; Zdrojewski, T.; et al. Relationship between diabetes mellitus and atrial fibrillation prevalence in the Polish population: a report from the Non-invasive Monitoring for Early Detection of Atrial Fibrillation (NOMED-AF) prospective cross-sectional observational study. Cardiovasc Diabetol 2021, 20, 128, doi:10.1186/s12933-021-01318-2.
- Faul, F.; Erdfelder, E.; Lang, A.G.; Buchner, A. G*Power 3: a flexible statistical power analysis program for the social, behavioral, and biomedical sciences. Behav Res Methods 2007, 39, 175-191, doi:10.3758/bf03193146.
- Yao, Y.; Zhang, M.J.; Wang, W.; Zhuang, Z.; He, R.; Ji, Y.; Knutson, K.A.; Norby, F.L.; Alonso, A.; Soliman, E.Z.; et al. Multimodal data integration to predict atrial fibrillation. Eur Heart J Digit Health 2025, 6, 126-136, doi:10.1093/ehjdh/ztae081.
- Schnabel, R.B.; Sullivan, L.M.; Levy, D.; Pencina, M.J.; Massaro, J.M.; D'Agostino, R.B., Sr.; Newton-Cheh, C.; Yamamoto, J.F.; Magnani, J.W.; Tadros, T.M.; et al. Development of a risk score for atrial fibrillation (Framingham Heart Study): a community-based cohort study. Lancet 2009, 373, 739-745, doi:10.1016/s0140-6736(09)60443-8.
- Chamberlain, A.M.; Agarwal, S.K.; Folsom, A.R.; Soliman, E.Z.; Chambless, L.E.; Crow, R.; Ambrose, M.; Alonso, A. A clinical risk score for atrial fibrillation in a biracial prospective cohort (from the Atherosclerosis Risk in Communities [ARIC] study). Am J Cardiol 2011, 107, 85-91, doi:10.1016/j.amjcard.2010.08.049.
- Alonso, A.; Krijthe, B.P.; Aspelund, T.; Stepas, K.A.; Pencina, M.J.; Moser, C.B.; Sinner, M.F.; Sotoodehnia, N.; Fontes, J.D.; Janssens, A.C.; et al. Simple risk model predicts incidence of atrial fibrillation in a racially and geographically diverse population: the CHARGE-AF consortium. J Am Heart Assoc 2013, 2, e000102, doi:10.1161/jaha.112.000102.
- Wu, J.; Nadarajah, R.; Nakao, Y.M.; Nakao, K.; Arbel, R.; Haim, M.; Zahger, D.; Lip, G.Y.H.; Cowan, J.C.; Gale, C.P. Risk calculator for incident atrial fibrillation across a range of prediction horizons. Am Heart J 2024, 272, 1-10, doi:10.1016/j.ahj.2024.03.001.
- Pastori, D.; Menichelli, D.; Li, Y.G.; Brogi, T.; Biccirè, F.G.; Pignatelli, P.; Farcomeni, A.; Lip, G.Y.H. Usefulness of the C(2)HEST score to predict new onset atrial fibrillation. A systematic review and meta-analysis on >11 million subjects. Eur J Clin Invest 2024, 54, e14293, doi:10.1111/eci.14293.
- Joglar, J.A.; Chung, M.K.; Armbruster, A.L.; Benjamin, E.J.; Chyou, J.Y.; Cronin, E.M.; Deswal, A.; Eckhardt, L.L.; Goldberger, Z.D.; Gopinathannair, R.; et al. 2023 ACC/AHA/ACCP/HRS Guideline for the Diagnosis and Management of Atrial Fibrillation: A Report of the American College of Cardiology/American Heart Association Joint Committee on Clinical Practice Guidelines. Circulation 2024, 149, e1-e156, doi:10.1161/cir.0000000000001193.
- Wagner, M.W.; Namdar, K.; Biswas, A.; Monah, S.; Khalvati, F.; Ertl-Wagner, B.B. Radiomics, machine learning, and artificial intelligence-what the neuroradiologist needs to know. Neuroradiology 2021, 63, 1957-1967, doi:10.1007/s00234-021-02813-9.

Round 2
Reviewer 2 Report
Comments and Suggestions for Authors
It can be accepted, author has done significant improvement.